# How and Why Chromosomes Interact with the Cytoskeleton during Meiosis

**DOI:** 10.3390/genes13050901

**Published:** 2022-05-18

**Authors:** Hyung Jun Kim, Chenshu Liu, Abby F. Dernburg

**Affiliations:** 1Department of Molecular and Cell Biology, University of California, Berkeley, CA 94720-3200, USA; hjunkim@berkeley.edu; 2Howard Hughes Medical Institute, 4000 Jones Bridge Road, Chevy Chase, MD 20815-6789, USA; chenshu.liu@berkeley.edu

**Keywords:** meiosis, chromosomes, LINC complex, cytoskeleton, nuclear envelope, homolog pairing, synapsis

## Abstract

During the early meiotic prophase, connections are established between chromosomes and cytoplasmic motors via a nuclear envelope bridge, known as a LINC (linker of nucleoskeleton and cytoskeleton) complex. These widely conserved links can promote both chromosome and nuclear motions. Studies in diverse organisms have illuminated the molecular architecture of these connections, but important questions remain regarding how they contribute to meiotic processes. Here, we summarize the current knowledge in the field, outline the challenges in studying these chromosome dynamics, and highlight distinctive features that have been characterized in major model systems.

## 1. Introduction

Segregation of homologous chromosomes during meiosis is essential for sexual reproduction. During the first meiotic division (Meiosis I), homologous chromosomes segregate from each other to generate haploid gametes. Faithful partitioning of homologs depends on their prior pairing, synapsis, and recombination during meiotic prophase.

The processes of pairing and synapsis are facilitated by active movements of chromosomes within the nucleus. These are mediated by connections between specific chromosome loci and cytoskeletal motors outside of the nucleus, which are usually established upon meiotic entry and require transmembrane proteins that span the intact nuclear envelope (NE). In some cases, the nucleus also rotates, oscillates, or translocates within the cell volume [1,2,3,4]. They can lead to the clustering of chromosome regions in a region of the nuclear envelope near the spindle polar body (SPB) or centrosome, resulting in a chromosome configuration called the “meiotic bouquet” [5,6,7,8,9]. These movements typically abate once chromosomes are fully paired and synapsed.

Telomeres, the physical ends of linear eukaryotic chromosomes, are the most common loci that mediate attachment to the NE, but analogous roles are played by centromeric regions in *Drosophila* and specialized meiotic “Pairing Centers” in the nematode *Caenorhabditis elegans* [3,10,11,12]. In the ciliate *Tetrahymena thermophila*, centromeres and telomeres are both tethered and form clusters at opposite ends of the nucleus [13]. While the chromosome loci that mediate attachment vary, in most or all organisms, these connections depend on LINC (linker of nucleoskeleton and cytoskeleton) complexes, comprised of pairs of SUN (Sad1 and UNC-84 homology) and KASH (Klarsicht, ANC-1, and Syne-1 homology) domain proteins that span the inner and outer nuclear membranes, respectively, and interact within the perinuclear lumen—the space between the nuclear membranes [9,14]. The amino termini of SUN domain proteins are typically intrinsically disordered and extend into the nucleus to interact with proteins bound to telomeres or other chromosome regions, while carboxy termini containing the SUN domain reside in the lumen, where they interact with the KASH domains. The amino termini of KASH domain proteins protrude into the cytosol and typically interact with cytoskeletal filaments or motors [9,15].

Although NE attachment and movement of chromosomes during meiosis are widely conserved across eukaryotes, the key functions of these chromosome dynamics are still under debate [15]. A longstanding hypothesis is that associations between chromosomes and the NE may promote homology search by reducing its dimensionality from the 3D nuclear volume to the 2D nuclear surface. Some evidence directly supports the role of attachment and/or movement in accelerating homolog pairing [16,17,18,19]. Subtelomeric sequences play key roles in recombination partner choice in some organisms, consistent with the idea that telomere-led movements promote homology search within adjacent regions [20,21,22]. However, homolog pairing is observed even prior to chromosome association with LINC complexes in some organisms, and active chromosome movements often persist until, or even after, synapsis is completed, suggesting that these interactions may also play other roles [6,23]. In *C. elegans*, the disruption of these attachments leads to nonhomologous synapsis, indicating that they inhibit inappropriate pairing and/or SC formation, in addition to promoting proper pairing [10,12,15]. Both computational simulations and experimental evidence have indicated that chromosome attachments to LINC complexes and the resulting chromosome movements help to eliminate nonhomologous interactions and/or resolve chromosome entanglements [24,25,26]. They can also promote spreading of the SC along paired chromosomes from sites of nucleation [27,28]. 

Ambiguity about the functions of these movements stems, in part, from experimental challenges in studying these mechanisms, as well as from the apparent differences in the consequences of disrupting the components of these movement systems in diverse experimental models. Some of the proteins involved, including the LINC complexes and cytoskeletal filaments and motors, play other essential roles in cell proliferation and development. Additionally, it is challenging to directly observe and quantify meiotic chromosome dynamics, which can last from several hours to days. Meiotic cells can also display indirect consequences of disrupting these movements, often reflecting the activation of meiotic checkpoints that monitor the synapsis, induction and repair of double-strand breaks (DSBs), or formation of crossover precursors, and they feed back to regulate meiotic progression. Thus, defects in attachment or movement may lead to cell cycle delays, changes in the number and distribution of DSBs, altered crossover patterning, and/or apoptosis of meiotic cells, confounding the analysis of the direct effects of these movements. The impact of these surveillance mechanisms varies markedly between organisms [29,30]. In most cases, chromosome connections and movements are essential for the successful completion of meiosis, while, in others, particularly in budding yeast, the effects of disrupting these processes are quite subtle [31,32,33,34].

Here, we review how early meiotic events, including chromosome reorganization, homolog pairing, and meiotic recombination, are coordinated with chromosome–LINC complex cytoskeletal interactions, highlighting the conserved and divergent features among the experimental organisms. We also summarize the current knowledge regarding the molecular requirements for chromosome attachment and movement in major experimental models. 

### 1.1. Homolog Pairing during Meiosis

Chromosome attachment to and movement along the NE typically occur during early meiotic prophase, as chromosomes first establish physical contact with their homologous partners [6,9]. In most eukaryotes, the process of homolog pairing is coupled with assembly of the synaptonemal complex (SC), although some organisms have lost the ability to form this structure. SC assembly initiates at discrete points between each chromosome pair and extends processively from these sites, thereby “zippering” homologs into side-by-side alignment along their lengths [6,35]. 

In some species, including budding yeast and mammals, the nucleation of the synapsis depends on and occurs at sites where a DSB has been made and initial steps in homologous recombination, including inter-homolog strand invasion, have occurred [36]. Proximity to a chromosome attachment site likely facilitates the homology search required for strand invasion; this may account for the tendency of synapsis to initiate in subtelomeric regions in many species [16,20,21,22]. DSBs may also be enriched in these regions, perhaps as a mechanism to promote efficient pairing and synapsis [22].

In a few organisms, homologous synapsis occurs even in the absence of DSB induction. These include the fruit fly *Drosophila melanogaster* and nematode *Caenorhabditis elegans*, which have independently derived recombination-independent pathways for pairing and synapsis [6,37]. In *Drosophila* females, centromeric regions first cluster near the nuclear envelope and then pair in premeiotic cells [38,39]. Homologs probably separate during the intervening mitotic divisions, so it is unclear how this premeiotic pairing contributes to homologous synapsis during meiotic prophase; it may simply be that mechanisms that promote pairing during meiotic prophase are initiated prior to meiotic entry.

### 1.2. Nuclear Envelope Remodeling

The nuclear envelope in eukaryotes comprises of the inner and outer nuclear membranes (the latter of which is contiguous with the endoplasmic reticulum), nuclear pores that penetrate both membrane layers, and numerous additional membrane-spanning and -associated proteins. A fibrous protein network often assembles along the interior surface of the inner membrane, providing mechanical support. In metazoans, this nucleoskeleton is comprised of intermediate filament proteins known as lamins. Although lamin homologs are not found outside of metazoans [40], similar structures have been observed in protists and plants, and coiled-coil proteins that contribute to this structure have been identified in plants [41,42,43]. 

The nuclear lamina creates a barrier to chromosome diffusion in many cells, since it behaves as a stable meshwork to which chromosomes are physically tethered. Large-scale movement of chromosomes along the nuclear envelope during meiosis probably requires remodeling of the lamina [44]. The mechanisms by which this occurs vary widely among metazoans: the lamina is reduced or absent during early meiosis in some species, such as chickens [45], while, in others, meiotic cells express specialized lamin isoforms that form a more dynamic network [44,46]. The lamina can also be remodeled through posttranslational modifications that reduce protein–protein binding, analogous to and likely overlapping with the mechanisms that promote lamina disassembly during nuclear divisions [47,48].

### 1.3. Chromosome–Nuclear Envelope Attachments during Meiosis

In most organisms studied to date, the meiotic connections between the chromosomes and cytoskeleton are mediated by LINC complexes, which are comprised of SUN and KASH domain proteins. These proteins are more broadly conserved than lamins and have diversified within some clades to form large families [15,49]. Homologs of these proteins have not been detected in ciliates but may simply have diverged beyond recognition. 

In some cases, two or more SUN domain proteins contribute to meiotic chromosome movements, and loss of one results in only partial defects [50,51,52]. LINC proteins that contribute to meiotic chromosome movement typically play other essential roles, e.g., as components of the spindle pole body in fungi, as well as the links between the centrosomes, nucleus, and mediators of nuclear positioning and movement in metazoans [49,53,54]. Thus, they are typically broadly expressed, but some may be restricted to meiosis—e.g., budding yeast Csm4 is a meiosis-specific paralog of Msp2 [34].

Meiotic chromosome attachments and movements often depend on additional inner nuclear membrane proteins, expressed specifically during meiosis. These include Bqt3 and Bqt4 in fission yeast and MAJIN (membrane-anchored junction protein) in most metazoans [55,56,57]. How these transmembrane proteins contribute to meiotic attachment and movement remains unclear; they may serve as adaptors to connect SUN proteins to chromatin-binding proteins and/or promote other activities of the LINC complexes, such as clustering or force transduction.

SUN domains form homo- or heterotrimers [58]; these trimers may dimerize to form symmetrical hexamers, which have been proposed to lead to extended branched networks (Figure 1) [59]; however, it is unclear whether such interactions occur in vivo. If so, extended networks may contribute to the clustering of LINC proteins during meiosis, as well as to their ability to sustain and/or respond to forces acting tangential to the plane of the NE. However, deletion of the trimeric coiled-coil region of *C.*
*elegans* SUN-1 does not impair meiosis, suggesting that if such networks indeed promote clustering of LINC complexes in meiosis, they do not depend on this coiled-coil region [60]. Alternatively, meiosis-specific transmembrane proteins, such as MAJIN and Bqt3/4, may promote associations between LINC proteins that lead to clustering.

Chromosome attachment to LINC complexes also typically requires expression of meiosis-specific proteins that bind to chromosomes. These include Ndj1 (*S. cerevisiae*), Bqt1/2 (*S. pombe*), TERB1/2 (most metazoans), and the pairing center proteins HIM-8 and ZIM-1, -2, -and -3 (*C. elegans*) [31,32,56,57,61,62,63,64]. Some of these have DNA-interacting domains, while others interact with constitutively expressed DNA-binding proteins, such as Rap1 (yeast) or TRF1 (vertebrates). In budding yeast, the incorporation of the histone variant H2A.Z at telomeres during meiosis is also important for their interaction with the SUN domain protein Mps3 [65,66].

In addition to the structural components of the chromosome–NE linkage complexes, regulatory kinases are often concentrated at these sites. In *C. elegans*, two meiotic kinases, CHK-2 and PLK-2, are recruited by the HIM-8 and ZIM proteins, which contain zinc finger domains that recognize the DNA sequence motifs enriched in the pairing center regions [67,68,69]. In mice, SUN1 interacts with CDK2 during meiosis, likely by binding directly to a cyclin-like protein, Speedy A, that partners with CDK2 at these sites [70,71,72]. In fission yeast, telomeres recruit Cdk1 and promote the accumulation of the kinase with the spindle pole body, which eventually leads to the exit from the bouquet stage and nuclear division [73,74]. 

In some cases, these kinases are thought to directly promote chromosome attachment or movement, e.g., by modifying the LINC proteins and/or lamina [11,47,68,72,75]. Kinase activity may also contribute to regulating synapsis initiation and/or may be components of checkpoints that monitor synapsis [67,68,69,71]. Intriguingly, some kinases associated with attachment sites also play roles in crossover formation [76,77]. Thus, chromosome–LINC complex attachments not only promote pairing and synapsis, but often act as regulatory hubs that control meiotic progression.

Below, we highlight features of these attachment complexes specific to each of several major model systems and briefly summarize the reported consequences of disrupting these interactions. 

#### 1.3.1. *Mus musculus* (Mouse)

In mice, meiosis-specific proteins TERB1 and TERB2 are required to connect the telomere-associated protein TRF1 to LINC complexes [56,61]. This interaction requires the small inner NE protein MAJIN [56]. Homologs of TERB1, -2, and MAJIN are detected in most metazoans, suggesting conserved meiotic telomere-LINC complex interactions [57]. The meiotic LINC complex is comprised of SUN1 and KASH5 [17,78]. SUN2 also colocalizes with telomeres [52] but presumably plays a minor role, since its absence does not impair meiosis, in contrast to SUN1. The cytosolic domain of KASH5 interacts with dynein to promote movement along the microtubules [78,79]. SUN1 interacts with Speedy A, a cyclin-like protein that recruits and activates CDK2 [70]; this activity is required for the tethering of telomeres to LINC complexes [71,72]. Nuclear rotation and chromosome movements occur concomitantly during meiotic prophase in mouse spermatocytes [4]. Telomere movements are reduced in recombination and synapsis mutants, perhaps due to feedback regulation by other meiotic checkpoints [4]. 

In *Terb1*, *Terb2*, *Majin*, and *Sun1* mutants, telomeres fail to attach to LINC complexes, homolog pairing is disrupted, most chromosomes remain unsynapsed, and the cell cycle is arrested [17,56,61,80]. In contrast, mutations in *Cdk2* result in extensive nonhomologous synapsis and partner switching, indicating that Cdk2 is dispensable for synapsis initiation, but important for its proper regulation [71,81].

#### 1.3.2. *Danio rerio* (Zebrafish)

The homologs of mouse TERB1, TERB2, and MAJIN are expressed in zebrafish meiosis, suggesting that a similar molecular mechanism connects the telomeres to LINC complexes [57]. Microtubules concentrate adjacent to the attachment sites [82], and microtubule depolymerization disrupts bouquet formation [83]. Rapid chromosomal rotations have been observed through live imaging [84]. The consequences of disrupting the bouquet have not been analyzed.

Pairing of telomeres and subtelomeric regions can be detected before synapsis, and synapsis often initiates near clustered telomeres [22]. Interestingly, DSBs markers are also enriched in subtelomeric regions [22].

#### 1.3.3. *Caenorhabditis elegans*

In *C. elegans*, regions known as homolog recognition regions (HRRs) or pairing centers (PCs) lie near one end of each of the six chromosomes. These regions span up to a megabase and contain numerous (~100–1000) dispersed binding sites for one of four paralogous zinc finger (ZnF) proteins: ZIM-1, 2, 3, and HIM-8 [63,64,85,86]. These proteins contain short linear motifs in their unstructured N-termini that recruit the meiotic kinases CHK-2 and PLK-2 [67,68,69]. The activity of both kinases is required to mediate the interaction of PCs and/or clustering of the LINC proteins SUN-1 and ZYG-12 [68,69]. The cytosolic domain of the KASH protein ZYG-12 interacts with dynein and generates the processive movements of chromosomes along the microtubules, which accelerate pairing and are important for synapsis [12,87]. Cytoplasmic centrosomes are inactivated during meiotic prophase; consequently, a discrete microtubule aster and tight clustering of PCs are absent, but the nuclei take on a polarized appearance, due in part to the displacement of the nucleolus towards one side of the nucleus [88].

The deletion of PCs or mutation of the ZnF proteins disrupts the attachment between chromosomes and LINC complexes and abrogates pairing and synapsis of the corresponding chromosomes [63,64,86]. Deletion of all four ZnF proteins leads to dysregulated synapsis, perhaps because the PCs normally sequester the activities that promote synapsis initiation [68,69]. Absence or mutation of SUN-1 also results in nonhomologous synapsis [10,12], while mutations in or depletion of ZYG-12 or dynein results in aberrant and reduced synapsis [12,89]. Direct visualization of synapsis has confirmed that it usually initiates near the NE, and also revealed that chromosome movements promote SC elongation in *C. elegans*, possibly by facilitating homolog alignment and/or resolving entanglements [27]. 

#### 1.3.4. *Drosophila melanogaster*

In *Drosophila*, the SUN domain protein Klaroid and its KASH partner Klarsicht tether centromeres to NE and interact with microtubules, inducing nuclear rotations and centromere clustering and pairing [3]. This initiates in germline cells, prior to the last mitotic division before the onset of meiosis [38,39]. Despite the role of centromeres in these dynamics, specific pairing of homologs initiates in the euchromatic regions [38,39,90,91]. Meiosis in males lacks synapsis and crossing-over, but homolog pairing may be promoted by mechanisms that overlap with those in oocytes [92].

Mutations in either LINC protein result in defects in centromere pairing. Synapsis is greatly diminished in *klarsicht* (KASH) mutants. More extensive SC assembly is observed in *klaroid* (SUN) mutants, but it is not yet clear whether this occurs between homologs. Chromosome motion is also more strongly affected in *klarsicht* than in *klaroid* mutants, perhaps due to residual nuclear rotation resulting from connections between Klarsicht and dynein [3]. These movements may require an inner nuclear envelope protein that is partially redundant with Klaroid, but this has not been established.

#### 1.3.5. *Tetrahymena thermophila*

The ciliate *Tetrahymena thermophila* has two classes of nuclei: a somatic macronucleus (MAC) that contains many tiny chromosomes and diploid micronucleus (MIC) that can undergo mitosis or meiosis. In early meiosis, the MIC becomes polarized, with centromeres and telomeres clustered at opposite sides, and then elongates dramatically [93]. Loss of centromere clustering results in more severe pairing defects than disrupted telomere clustering [13,94]. 

This reorganization requires microtubules and depends on the formation of DSBs and the DNA-damage sensing kinase ATR [95,96]. Ectopic DNA damage (e.g., induced by UV or Cisplatin) restores MIC elongation, but not homolog pairing, in *spo11* mutant cells [95,96], indicating that the role of DSBs in homolog pairing and early meiotic nuclear reorganization can be uncoupled. This microtubule-dependent polarization resembles the bouquet stage in other eukaryotes, particularly fission yeast (see below) [13,95]. Notably, homologs of LINC complex proteins have not been identified in *Tetrahymena*, and nuclear envelope components required for this reorganization are currently unknown [93].

#### 1.3.6. *Saccharomyces cerevisiae* (Budding Yeast)

Budding yeast have an unusual mechanism of meiotic chromosome movement that involves actin-based motors, rather than microtubules. A meiosis-specific adaptor protein, Ndj1, connects the telomeres to the ubiquitously-expressed SUN domain protein Mps3 [31,32,33]. Csm4 and Mps2 are both KASH domain-like proteins; Csm4 expression is restricted to meiosis. Both paralogs participate in rapid chromosome movements by interacting with Myosin II, which moves along the actin filaments [34,97,98,99].

Mutations that impair the functions of Ndj1, Mps3, or Csm4 delay meiosis, reduce recombination, and impair spore viability [31,32,33,97]. Polycomplexes (self-assemblies of SC proteins) are also prevalent in the nucleoplasm of *ndj1* and *csm4* mutant cells, suggesting that telomere-led motion may promote SC assembly [28,100], as in *C. elegans*. Nevertheless, only a modest delay in pairing at individual chromosome loci is seen in *ndj1*, *mps3*, and *csm4* mutants [18,31,33,34,100]. Thus, telomere-LINC complex association and the resulting movements are not strictly required for homolog pairing and recombination. This may reflect the contributions of other homolog alignment mechanisms, as seen in premeiotic cells [6,23]. Alternatively/additionally, recombination-mediated homology search within the small yeast nucleus may be sufficient to pair chromosomes efficiently. 

#### 1.3.7. *Schizosaccharomyces pombe* (Fission Yeast)

Fission yeast Bqt1 and Bqt2 proteins connect telomeres to LINC complexes in the NE [62]. Bqt1 and Bqt2 connect the Rap1 and Taz1 that bind the telomeres to the SUN domain protein Sad1, which interacts with the KASH domain protein Kms1 [101,102,103]. Kms1 interacts with dynein motor protein to generate chromosome movements and an extended “horsetail” configuration of the nucleus [16,104,105]. Two additional NE proteins, Bqt3 and Bqt4, are required to “prime” the interaction of these proteins with Sad1 [55]. 

In the absence of Rap1, Taz1, Bqt1, Bqt2, or Bqt4, telomeres fail to connect to LINC complexes, leading to severely decreased spore viability [55,62,101,103]. Absence of the KASH protein Kms1 results in loss of telomere clusters, while it has only mild effects on spore viability [102]. Homolog pairing is disrupted upon removal of telomere-LINC complex attachment or chromosome movements, suggesting that these attachments are required for pairing [16]. Interestingly, the pairing of centromeric regions is more resistant to removal of Taz1 compared to chromosome arms, while the loss of chromosome movements impairs the pairing of both the arm and centromeric regions [16], which suggests that different mechanisms contribute to the pairing of chromosome arms vs. centromeric regions. 

#### 1.3.8. Higher Plants

A bouquet stage mediated by connections between telomeres and LINC complexes has been described in diverse crop plants, such as wheat, barley, rice, and maize, as well as in the model cress *Arabidopsis thaliana* [50,51,106,107]. Centromeres also tend to be polarized and clustered near the nuclear periphery, but this is insensitive to microtubule depolymerization, suggesting that it may be a byproduct of prior mitosis [106,107].

The SUN domain family has expanded in many plants. At least two paralogs contribute to telomere movement and clustering in *Arabidopsis*, maize, and rice [50,51,107]. In *Arabidopsis*, the loss of AtSUN1 and AtSUN2 perturbs meiosis, yet roughly half of the chromosomes still successfully pair and synapse [50]. Similar effects are observed in rice lacking OsSUN1 and OsSUN2 [51]. Thus, in these higher plants, LINC complexes are required for bouquet formation, while pairing and synapsis are promoted by other mechanisms, as in budding yeast.

## 2. Concluding Remarks

Functional analysis of chromosome attachment and movement during meiosis can be complicated by other essential roles for LINC complexes and cytoskeletal components in cell proliferation and development [53]. Thus, most studies of chromosome attachment and movement have focused on the analysis of meiosis-specific factors, but better tools for conditional gene inactivation and targeted protein depletion are enhancing the toolbox for investigation of meiosis and should help to clarify both the mechanisms and consequences of chromosome attachment and movement. 

Direct observation and quantification of meiotic chromosome movements in several experimental systems, including fission yeast, budding yeast, mouse spermatocytes, *Tetrahymena*, *C. elegans*, *Drosophila*, and some plants, has helped to clarify their effects on pairing, synapsis, and recombination. Such experiments are technically challenging, particularly in multicellular organisms, in which meiosis occurs in reproductive tissues that can be difficult to access, immobilize, and/or maintain in a fully functional state throughout imaging. Continual improvements in genome editing, fluorescent labeling, and microscopy methods will make such analyses more feasible and informative. By combining quantitative live imaging with molecular dissection and quantitative modeling, future work will likely reveal the functional impact of these unique meiotic chromosome dynamics in unprecedented detail.

## Figures and Tables

**Figure 1 genes-13-00901-f001:**
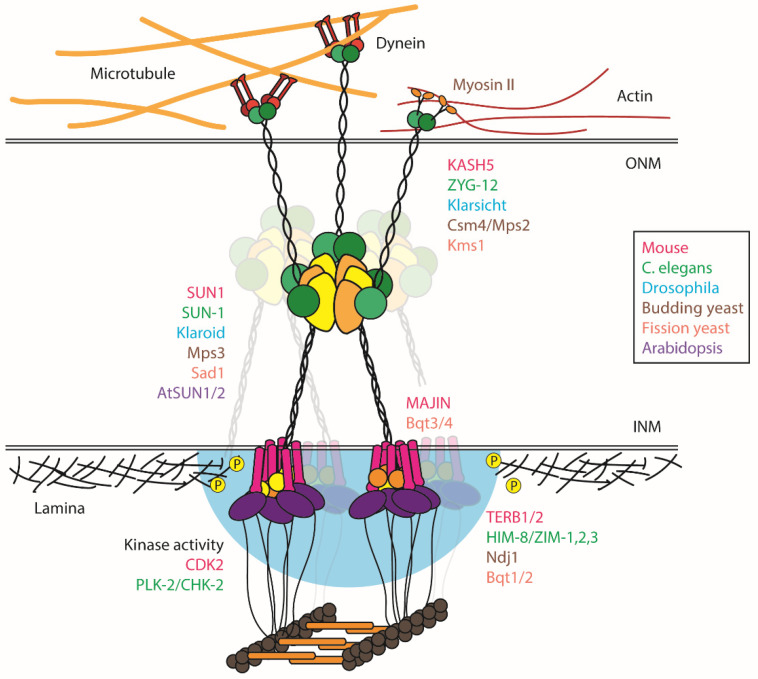
Schematic representation of meiotic chromosome–NE–cytoskeletal complexes in diverse species. The SUN domain forms a trimer, and an adjacent extended α-helical domain forms trimeric coiled-coils, while KASH domain proteins dimerize. SUN domain trimers can dimerize and are proposed to interact with KASH domains from different dimers to create branched complexes, which may contribute to the clustering of chromosome attachment sites during meiosis. The nucleoplasmic domain of SUN proteins interact with chromosomal proteins, which often depends on the expression of meiosis-specific adapter proteins. Small proteins that span the inner nuclear membrane, including MAJIN in most metazoans and Bqt3/4 in fission yeast, are also essential for the coupling of chromosomes to the LINC complex and/or clustering of LINC complexes (translucent structures in the background). Cytosolic domains of KASH domain proteins often interact with dynein or microtubules; however, in budding yeast, they link telomeres to Myosin II on actin filaments. CDK2 and PLK-2/CHK-2 kinases are recruited to chromosome–LINC complex attachment sites in mice and *C. elegans*, respectively, and their activities (blue cloud) are required for the connection of telomeres/Pairing Centers to LINC complexes. In *C. elegans*, phosphorylation of the nuclear lamina by PLK-2 liquefies or disrupts the nuclear lamina to promote chromosome movements. In mice, CDK2 activity also limits promiscuous synapsis between nonhomologous chromosomes.

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
