# Peer review of "How and Why Chromosomes Interact with the Cytoskeleton during Meiosis"

_genes, 2022, doi:10.3390/genes13050901_

Round 1
Reviewer 1 Report
This review by Kim and colleagues covers the earliest steps of meiosis, when paternal and maternal chromosomes move, assess their homology, align along their length and pair. Recent findings have shown that the cytoplasmic cytoskeleton and its links to the nuclear envelop (LINC) play a key role in this process. This topic has been reviewed several times already but the authors do a great job at comparing many species from animals to plants, while keeping the manuscript short. Interestingly, they also point to different severity of phenotypes in different species. This review is interesting, well-written and will be useful for many readers.
I only have minor concerns:
- Line 9: starting the first sentence of the abstract with the “bouquet” may be difficult to understand for non-specialist readers
- Line 21: “ homologous chromosomes separate from each other…” I would write “chromosome segregate..”
- Line 38 “ it seems likely that in all organisms these connections depend on LINC…” However, it is later written that homologues of LINC haven’t been identified in Tetrahymena Line 262.
- Line 99 : Joyce et al 2013 and Christophorou et al 2013 (ref 86 and 87) should be cited here.
- Line 254: the authors propose that klaroid weaker phenotype on chromosome pairing compared with klarsicht could be caused by redundancy. It is possible. However, in klaroid mutant germ cells, nuclei are still rotating because Klarsicht is probably still linked to microtubules, which is not the case in klarsicht mutant flies. It is thus also possible that greater dynamics in klaroid mutants leads to weaker phenotypes.
- The figure is nice. I would add the names of motor proteins (Dynein and Myosin). It could also be interesting to have different colors for different species. It is hard to remember in the list of homologues to which organism they belong.
Reviewer 2 Report
This review provides a summary of current knowledge in the field describing the mechanisms and roles of meiotic chromosome attachment to the nuclear envelope via the LINC complex and force-generated movements during meiosis. Distinctive features between different model organisms are discussed. In addition, the authors provide important insights into some of the major challenges of studying these processes, especially since many proteins play other roles in the biology of the cell.
- The manuscript would be improved if a table listing mitotic and meiotic paralogs of each protein within a species and their orthologs in different species was provided.
- Line 45. What is “still under debate” should be expanded. The rest of the paragraph focuses on how experimentally disrupting motion can affect other cellular processes which can confound interpretation of data, however, what aspect of these examples is under debate is not given. A separate paragraph covering “debatable” functions could be expanded to focus on whether motion has evolved to take apart nonspecific interactions, relief of interlocks and/or entanglements, or simply to agitate the nucleus to facilitate random interactions.
- Section 1.1. Given the importance of chromosome attachment to the NE and the generation of force are so integral to homolog pairing, this section could be expanded to give a more comprehensive overview of pairing. The authors might point out that there are several organisms where DSB-dependent pairing first appears at telomeres (humans, some plants and zebrafish) in contrast to mouse and yeast where they occur throughout the genome.
- Section 1.3 This section may also be a reasonable place to bring up the Rabl conformation, which has not been addressed in the review, yet is likely important in species such as pombe the bouquet represents an inversion of the Rabl configuration. This happens also to a lesser extent in S. cerevisiae.
- Lines 126-128 The LINC acronym is spelled out in the introduction and in this paragraph, yet SUN is only spelled out here. For consistency, they should be spelled out either in the introduction or here but not necessarily both.
- Section 1.3.2 Line 261, It has been shown that synapsis does not occur in a spo11 mutant in zebrafish [28], supporting the dependence of synapsis on DSBs.
- Section 1.3.6. Gene names should be italicized
- Line 307; insert “the” bouquet
- Figure 1. What is the significance of the less opaque structures in the background? What is the significance of the blue region? There seems to be no reference to this figure in the text.
